# Indole-3-Carbinol Inhibits *Citrobacter*
*rodentium* Infection through Multiple Pathways Including Reduction of Bacterial Adhesion and Enhancement of Cytotoxic T Cell Activity

**DOI:** 10.3390/nu12040917

**Published:** 2020-03-27

**Authors:** Yanbei Wu, Qiang He, Liangli Yu, Quynhchi Pham, Lumei Cheung, Young S. Kim, Thomas T. Y. Wang, Allen D. Smith

**Affiliations:** 1China-Canada Joint Lab of Food Nutrition and Health (Beijing), Beijing Technology & Business University, Beijing 100048, China; yanbeiwu@btbu.edu.cn; 2Beijing Advanced Innovation Center for Food Nutrition and Human Health, Beijing Technology & Business University, Beijing 100048, China; 3Diet, Genomics, and Immunology Laboratory, Beltsville Human Nutrition Research Center, USDA-ARS, Beltsville, MD 20705, USA; quynhchi.pham@usda.gov (Q.P.); lumei.cheung@ars.usda.gov (L.C.); 4College of Biomass Science and Engineering, Sichuan University, Chengdu 610065, China; heq361@163.com; 5Department of Nutrition and Food Science, University of Maryland, College Park, MD 20742, USA; lyu5@umd.edu; 6Nutritional Science Research Group, Division of Cancer Prevention, National Cancer Institute, National Institutes of Health, Bethesda, MD 20892, USA; kimyoung@mail.nih.gov

**Keywords:** indole-3-carbinol, *Citrobacter rodentium*, colitis, bacterial adhesion, cytotoxic T cells

## Abstract

Intestinal inflammation is associated with an increased risk of developing colorectal cancer and may result from dysregulated responses to commensal bacteria or exposure to bacterial pathogens. Dietary modulation of intestinal inflammation may protect against development of colon cancer. However, the precise diet-derived components and underlying mechanisms remain elusive. *Citrobacter rodentium* (*Cr*) induces acute intestinal inflammation and has been used to study the role of inflammation in the susceptibility to colon cancer. Here we examine the effects of indole-3-carbinol (I3C), a dietary compound with anticarcinogenic properties, on intestinal immune and inflammatory responses to *Cr* infection and adhesion to colonic cells in vitro. C57BL/6J mice were fed a diet with/without 1 μmol/g I3C and infected with *Cr*. Compared to infected mice fed with a control diet, consumption of a 1 μmol I3C/g diet significantly reduced fecal excretion of *Cr*, *Cr* colonization of the colon, and reduced colon crypt hyperplasia. Furthermore, expression of *Cr*-induced inflammatory markers such as IL-17A, IL-6, and IL1β were attenuated in infected mice fed with the I3C diet, compared to mice fed a control diet. The expression of cytotoxic T cell markers CD8 and FasL mRNA were increased in I3C-fed infected mice. In-vitro, I3C inhibited *Cr* growth and adhesion to Caco-2 cells. I3C alleviates *Cr*-induced murine colitis through multiple mechanisms including inhibition of *Cr* growth and adhesion to colonic cells in vitro and enhancement of cytotoxic T cell activity.

## 1. Introduction

Colorectal cancer (CRC) is the fourth most common malignancy and third leading cause of cancer deaths worldwide [1]. It is well documented that intestinal inflammation is one of the most common risk factors associated with CRC tumorigenesis, and is recognized as an important hallmark for CRC development [2,3]. Specific intestinal pathogens such as Streptococcus bovis/gallolyticus, as well as chronic intestinal inflammation caused by inflammatory bowel disease (IBD), were reported to contribute to CRC development [4,5]. Therefore, modulating inflammation in the intestine may influence/protect against development of CRC. Moreover, in most cases, CRC is thought to be caused by environmental factors since only 20–30% of CRC cases are attributed to familial hereditary [6]. Among the environmental factors, diet may be the most critical factor associated with CRC development [7]. Importantly, a diet rich in fruits and vegetables has been reported to exert anti-inflammatory effects in the intestine, and thus may help prevent development of CRC [8]. However, the precise dietary components and their modes of action (i.e., through modulation of inflammatory pathways) that contributed to CRC protective effects remain unclear and warrant further elucidation.

Indole-3-carbinol (I3C), a bioactive compound found in cruciferous vegetables such as broccoli, cabbage and cauliflower, is well known for its cancer protective effects [9,10]. When given orally through the diet or as a supplement, I3C can dimerize into another bio-active compound, 3,3′-diindolylmethane (DIM), under the acidic conditions in the stomach [11]. These indoles have been reported to exert various anti-bacterial, anti-inflammatory, and anti-cancer activities [12,13]. Several nuclear transcriptional factors including the estrogen receptor (ER), nuclear factor-κB (NF-κB), and the aryl hydrocarbon receptor (AhR) are modulated by I3C/DIM and contribute to the maintenance of hormonal homeostasis, inhibition of cell cycle progression/apoptosis, induction of DNA repair, and enhancement of carcinogen metabolism [14]. For example, dietary I3C was reported to regulate intestinal homeostasis via the formation of intestinal lymphoid follicles and the expansion of innate lymphoid cells through AhR-dependent pathways. I3C-derived DIM was found to alleviate oxazolone-induced colitis through Th2/Th17 suppression and T regulatory cell (Treg) induction [15]. These in vivo and in vitro findings support the value of I3C and its derivatives in modulating inflammation, cancer prevention and therapy.

The role of I3C and cruciferous vegetables in modulating pathogen-induced intestinal inflammation, via AhR-dependent signaling, has been reported for the *Citrobacter rodentium* (*Cr*) -induced colitis model [16,17]. *Cr* is a well-known murine mucosal pathogen that shares 67% of its genes with two clinically important human enteric pathogens: enteropathogenic *Escherichia coli* (EPEC) and enterohemorrhagic *E. coli* (EHEC) [18]. The *Cr* infection model shares many aspects of EPEC and EHEC infections in humans including the production of attaching and effacing lesions and the ability to promote colon tumorigenesis in mice [19]. *Cr* infection induces a variety of changes in the colon including hyperplasia and crypt dilation, epithelial cell proliferation, development of an uneven apical enterocyte surface, and mucosal thickening. Following oral administration, *Cr* initially colonizes in the caecum and then migrates to the colon with rapid expansion, the *Cr* excretion usually peaks on day seven post infection, and is gradually cleared by 21–28 days after infection. Both innate and adaptive immunity are important for the clearance of *Cr*. Resolution of *Cr* infection involves the recruitment of innate lymphoid cells, neutrophils, macrophages, B cells, T cells and the release of multiple cytokines and antimicrobial peptides [20]. CD4^+^ T cells, acting through IL-22, play a crucial role as a defense mechanism against *Cr* infection [21]. However, the T cell-derived cytokines (IL17A, IFN-γ, and TNF) are also known to, either directly or indirectly, contribute to intestinal tissue injury [22]. In addition to T-cells, the group 3 innate lymphoid cells (ILC3s) are also an important source of IL22 and contribute to clearance of *Cr*. AhR-dependent signaling pathways were also reported to be important in controlling and clearing *Cr* [23]. The effects of AhR in maintaining intestinal homeostasis were ablated in AhR-deficient mice and resulted in increased susceptibility of mice to *Cr* infection [15]. However, little information is available relating to the effect of I3C/DIM on bacterial growth and adhesion during *Cr* infection. Furthermore, the effect of I3C/DIM on CD8^+^ cells during *Cr* infection are also not well studied.

To address these questions, we characterized the effects of dietary I3C on the immune response to *Cr* infection and examined the in vitro effects of I3C on *Cr* growth and adhesion to a colonic cell line. We found the protective effect of dietary I3C against *Cr* infection may be through multiple mechanisms that involve modulation of bacterial growth and adhesion as well as the immune response to infection.

## 2. Materials and Methods

### 2.1. Animals and Diet

C57BL/6J mice (5-week-old male) were purchased from the Small Animals Division of the National Cancer Institute (Frederick, MD), and were housed in groups of four in ventilated cages with filter-tops at the Beltsville Human Nutrition Center‘s animal facility under a 12-h light cycle. To avoid possible sex effects on the infection, male mice were chosen. All experiments were approved by the USDA/ARS Beltsville Institutional Animal Care and Use Committee (protocol #12-030). Mice were acclimated for one week prior to the onset of dietary treatments. The mice were then randomized into two experimental feeding groups: (i) AIN-93M control diet, (ii) AIN-93M with 1 µmol (147.2 μg) I3C/g diet. The mice were fed the experimental diets for two weeks prior to infection with *Cr* and for the remainder of the experiment. Food consumption and body weights were recorded weekly.

It has been estimated that a dose of 1 µmol I3C/g diet (147 mg/kg) in mice equates to a human equivalent dose of 11.4 mg/ kg in the average adult human. If estimating this dosage for a 60 kg man, the amount of I3C provided in this study may correspond to the consumption of approximately half a head of cabbage. In a clinical study, administration of I3C supplements was tolerated in doses up to 1200 mg/ day in male and female cancer patients. Our selected concentration of I3C for this study is similar to those amounts consumed in the diet as well as those amounts supported on a low-dose chemopreventive range [24,25,26].

### 2.2. Cr Infection

The *Cr* strain used in this study is a nalidixic acid-resistant mutant of strain DBS100 (ATCC 51459). *Cr* was streaked out from a frozen stock on a Luria-Bertani (LB) agar plate and grown overnight at 37 °C. A colony was picked and used to inoculate LB medium and grown at 37 °C overnight. The following morning the culture was expanded, grown to an OD600 nm of ≈1.5, collected by centrifugation, and re-suspended in LB medium to a concentration of 5 × 10^10^ CFU/mL. Mice were infected by oral gavage with 0.2 mL of the *Cr* suspension (1.0 × 10^10^ CFU) or LB medium only for uninfected mice on either diet. The dose of oral gavage was determined by retrospective plating on LB agar plates containing 50 µg/mL nalidixic acid.

### 2.3. Sample Collection

Fresh fecal pellets of mice were collected on various days post infection (4, 7, 11, 14, 17, 20). The pellets were homogenized in LB broth, serially diluted, plated on LB/agar plates containing 50 µg/mL nalidixic acid, and incubated at 37 °C overnight. The colonies were enumerated the following day. Mice were sacrificed on day 12 and 21 after infection. The spleen and caecum were removed and weighed. The distal 5 cm of colon was excised and fecal pellets were removed and weighed. One cm portions of the colon were fixed in 4% formalin for histology or snap frozen in liquid nitrogen for gene expression analysis. To measure the degree of *Cr* colonization on the colon, the remaining colon tissue was weighed, homogenized, and plated on LB/agar plates with 50 µg/mL nalidixic acid. Results are presented as CFUs per gram of feces or colon.

### 2.4. Measurement of Crypt Depth

Approximately 1 cm portions of distal colon cut from equidistant positions were fixed in 4% formalin and embedded in paraffin before 5-μm sections were cut and stained with hematoxylin and eosin. Colon crypt depth was measured using a Nikon Eclipse E800 microscope (Nikon, Tokyo, Japan) and Nikon NIS-Elements software V4.6 (Nikon, Tokyo, Japan). The mucosal height for each mouse was determined by averaging 12 or more individual well-oriented crypts.

### 2.5. Measurement of Colonic Cytokine Gene Expression

To analyze gene expression in colon samples, total RNA was isolated using the TRIzol reagent (Life Technology, NY, USA). The concentration and integrity of RNA were measured using a Bioanalyzer (Agilent 2100 Bioanalyzer, Santa Clara, CA, USA). RNA with an integrity number above 8 was used for real-time qRT-PCR. The AffinityScript Multi-temperature cDNA Synthesis kit from Agilent was used to reverse-transcribe mRNA to complementary DNA. Real-time PCR was performed on Applied Biosystems ViiA7 Real-Time PCR System using TaqMan^®^ Gene Expression Assay (Invitrogen, Carlsbad, CA, USA). For evaluation of the treatment effects, genes of interest (Table 1) were normalized to the housekeeping gene TATA-box binding protein (TBP) and analyzed using the ∆∆Ct method. The primers/probes for gene expression analysis below were purchased from Life Technology.

### 2.6. Antimicrobial Susceptibility Assay

To determine the effects of I3C on *Cr* growth, *Cr* was cultured in Luria-Bertani (LB) broth overnight at 37 °C and the bacterial suspension was diluted to 2 × 10^7^ CFU/mL. One mL of the *Cr* suspension was added to wells of 24-well microtiter plates. Next, I3C (Sigma Chemical Co., St. Louis, MO, USA) was dissolved in ethanol and diluted to 0.5 mM or 1.0 mM in LB medium. For control wells not receiving I3C, LB containing an equivalent amount of ethanol was used. The concentrations of I3C were based on the minimum inhibitory concentration of I3C against the gram-positive bacterial strains previously evaluated by Sung et al [12]. The plates were incubated at 37 °C while on shaker (200 rpm) under aerobic and anaerobic conditions (BD GasPak EZ Anaerobe Gas Generating Pouch System with an indicator). The OD value was determined every hour using a microtiter plate reader (Microbiology International, Frederick, MD, USA).

### 2.7. Adhesion Assay

The method used to determine adhesion of *Cr* to Caco-2 cells was a modified procedure described by Cho et al. [27]. Briefly, human colonic Caco-2 cells (ATCC HTB-37) were grown in Dulbecco’s Modified Eagle medium (DMEM) supplemented with 20% (*v*/*v*) heat-inactivated fetal bovine serum, 2 mM L-glutamine, and 1% (*v*/*v*) antibiotics (100 U/mL penicillin, 100 μg/mL streptomycin). Cells were grown in 75 cm^2^ tissue culture flasks and incubated at 37 °C in 5% CO_2_. For consistency, cells at the 3rd passage after thawing from frozen stocks were used. For the adhesion assays, Caco-2 cells were seeded at a density of 1 × 10^5^ cells/well into 24-well microplates and incubated for 2 days to allow formation of a monolayer. One day prior to the assay, the complete medium was replaced with DMEM medium without the supplements. On the day of the assay, the medium was aspirated and replaced with 1 mL of bacteria resuspended at 1 × 10^6^ CFU/mL in freshly, non-supplemented DMEM medium containing various concentrations of I3C or DIM. The plates were then incubated at 37 °C for 1 h followed by four washes with PBS minus Ca^2+^/Mg^2+^ and the cells were lysed using 1 mL of cold 0.1% Triton X-100 in pure water. The resulting lysates were diluted (1:10) with PBS, plated on LB agar plates containing 50 µg/mL nalidixic acid, and cultured at 37 °C overnight. The colonies were enumerated the following day.

### 2.8. Statistcal Analysis

Data for each experimental group are expressed as the mean ± SD. Statistical analysis was performed using GraphPad Prism 7 (2018, GraphPad Software, San Diego, CA, USA). Significant differences in the means were detected using an unpaired Student’s t test (Mann–Whitney test) or one-way ANOVA followed by Fisher’s LSD test. Statistical significance was defined at *p* ≤ 0.05.

## 3. Results

### 3.1. Effect of Cr Infection and I3C Supplement on Food Intake and Body Weight

Appendix A shows the temporal changes of body weight of uninfected and infected C57/BL mice fed the control or the I3C diet. There were no significant differences in body weight between the four experimental groups. Appendix A illustrates food intake throughout the study. Although there’s a significant drop in food intake after infection, it was recovered by the end of the experiment, and there were no significant differences in food intake between the four experimental groups.

### 3.2. Effects of Dietary I3C on Cr Colonization in Feces and Colon

After the oral gavage with *Cr*, the bacteria initially colonize the caecum, and then are found in significant numbers in the colon by day three post infection [28]. Excretion of Cr peaked on day seven post infection. On day 20 post infection, the number of Cr in the feces of the infected mice were low, and there is no significant difference in fecal Cr excretion between the infected mice fed with the control and the I3C diet (Appendix A). Fecal Cr excretion was reduced in I3C-treated mice on day 11 post infection (Figure 1A). Moreover, the colonic tissue *C. rodentium* loads in the mice fed with the I3C diet were significantly lower than that in the mice fed by the control diet on day 12 post infection (Figure 1B).

### 3.3. Effects of Dietary I3C on Colon, Spleen and Cecum Weight in Mice

The effect of *Cr* infection and I3C treatment on the tissue/body weight ratio of colon, spleen, and caecum are shown in Figure 2A (day 12 post infection) and Figure 2B (day 21 post infection). As expected, colon weight increased in *Cr*-infected mice. Furthermore, I3C treatment had no effect on the infection-induced increase in colon weight. In addition, no difference was found in spleen or cecum weight between the four treatment groups both on day 12 and day 21 post infection.

### 3.4. Effects of Dietary I3C on Histologic Changes in Colon of Mice

Colon tissues obtained on day 12 and 21 were processed for H&E staining (Appendix A) and the mucosa height (bottom of crypts to epithelium) was measured. Infected mice had increased mucosa height compared to uninfected mice on both days 12 and 21 post infection but mice fed the I3C diet had reduced mucosal height (Figure 3) on day 12 post infection in comparison with the infected mice fed the control diet. However, there was no significant reduction in the crypt height between infected control- and I3C-fed mice on day 21.

### 3.5. Effects of Dietary I3C on AhR and Immune Markers in Colonic Tissues.

In agreement with previous studies, consumption of 1 µmol I3C/g diet significantly induced the expression of AhR-responsive genes including cytochrome P450 1A1 (CYP1A1) and 1B1 (CYP1B1) mRNA in colonic tissues. As compared to the control group, the expression of CYP1A1 and CYP1B1 mRNA was about 93 and 5 times higher in the I3C group, respectively (Figure 4). *Cr* infection had no effect on the expression of either CYP1A1 or CYP1B1 mRNA in mice fed the control diet but decreased expression of CYP1A1 mRNA in the mice fed the I3C diet (Figure 4).

As shown in Figure 5, mice infected with *Cr* had increased expression of multiple cytokines including IL-17A, IL-22, IL-6, and IL-1β mRNA in colonic tissue 12 days after infection. *Cr*-induced expression of IL-17A, IL-6, and IL-1β mRNA were significantly attenuated in infected mice that were fed on the I3C diet. However, there was no effect by I3C on a *Cr*-induced increase of IL-22 mRNA.

In addition to inflammatory marker genes closely associated with *Cr*-induced innate immunity, marker genes for the cytotoxic T cell and Treg cell were also determined. As illustrated in Figure 5, the expression of cytotoxic T cell markers CD8a, CD8b, and FasL mRNA were upregulated by the infection with *Cr* in colonic tissue. Importantly, I3C-fed *Cr*-infected mice have enhanced expression of all the aforementioned genes in the colonic tissue compared to *Cr*-infected mice fed control diet. Furthermore, gene expression of Foxp3, a transcriptional factor and specific marker of T regulatory cells that produces IL-10 [29], was significantly increased in the infected mice fed with the I3C diet in the colonic tissue (Figure 5). Consistent with this, mice fed with the I3C diet also had enhanced *Cr*-induced IL-10 mRNA expression in the colonic tissue of infected mice (Figure 5).

### 3.6. I3C Inhibited C. rodentium Growth

I3C has been reported to exhibit broad spectrum anti-microbial properties in vitro [12]. To test whether I3C can inhibit growth of *Cr* directly, the in vitro effect of I3C on *Cr* growth was investigated in both aerobic and anaerobic conditions. The concentration of I3C applied in this study was based on the reported minimum inhibitory concentration (40 μg/mL) of I3C for *Escherichia coli* [12]. I3C significantly inhibited *Cr* growth under anaerobic conditions in a dose-dependent manner (Figure 6) compared to the vehicle control (Ethanol). Under aerobic conditions, growth only appeared to be delayed as the growth rates were nearly identical (as indicated by parallel growth curves) starting at about 4 h.

### 3.7. I3C- and DIM-Inhibited Cr Adhesion to Caco-2 Cells

The effects of I3C (0–100 μM) and DIM (0–25 μM) on *Cr* adhesion to intestinal epithelial cells were also evaluated using a Caco-2 cell adhesion model. The concentrations of I3C and DIM used were not toxic to the Caco-2 cells. Both I3C and DIM treatment led to a dose-dependent inhibition of *Cr* adhesion to Caco-2 cells (Figure 7). Significant inhibition of bacterial adhesion was observed at 25 µM I3C and above, with an IC50 value of 35.2 μM. For DIM, significant inhibition was observed at 10 μM DIM and above, with an IC50 of 18.5 μM.

## 4. Discussion

The current study identified several novel observations related to the effects of I3C on *Cr* infection, specifically: (1) I3C or its bioactive derivative DIM can affect *Cr* growth and adhesion to intestinal cells in vitro; (2) dietary consumption of I3C enhanced cytotoxic T cell and Treg cell-related markers upon *Cr* infection; and (3) reduced colonic hyperplasia and the pro-inflammatory cytokine response. These data support the notion that dietary I3C or cruciferous vegetables may reduce intestinal infection-induced inflammation, a major risk factor for CRC. We postulate I3C may exert its protective effects through multiple mechanisms that include affecting bacterial growth and adhesion, as well as the immune response mediated by AhR. A working model summarized our observation is illustrated in Figure 8.

Most studies related to the protective effects of I3C on *Cr* infection have focused on modulation of the host immune responses by I3C. However, our result supports the idea that I3C may also be modulating *Cr* growth and adhesion to intestinal tissue in vitro. Firstly, I3C/DIM inhibits *Cr* growth under anaerobic conditions and secondly, I3C/DIM can prevent adhesion of *Cr* to intestinal epithelial cells in vitro. This notion is further supported by observing attenuated *Cr* loads in feces and colon of mice fed with a I3C diet (Figure 1). Lower bacterial loads would lead to lower immune responses and therefore the observed effects of I3C on attenuation of *Cr*-induced IL-1β, IL-6, and IL-17A levels could also be the result of a reduced *Cr* load. Alternatively, IL-10 and FoxP3 expression were upregulated in infected I3C mice compared to infected controls, and may have contributed to the decreased pro-inflammatory response. The mechanisms underlying the protective effects of I3C on *Cr* infection may be multiple. Recent studies indicated that I3C can modulate host immune responses as an AhR ligand and afford protective effects against *Cr* infection [17,30,31]. Signaling through the AhR receptor was reported to be a requirement for clearance of *Cr* infections. AhR knock-out mice are susceptible to lethal infection by *Cr* [16]. Furthermore, AhR ligand-deficient mice have increased susceptibility to lethal infection by *Cr*, but this can be reversed by administering I3C [32]. Our results showing that I3C induced an increase in CYP1A1 and CYP1B1 mRNA, two well-documented AhR-responsive genes, confirm that the AhR pathway is indeed activated in the intestinal tissues of mice, which may include intestinal cells as well as inflammation-related immune cells. Therefore, consistent with reported literature, activation of an AhR-mediated host immune system is an additional mechanism by which I3C may exert its *Cr* infection/inflammation protective effects. This is consistent with the downregulation of several pro-inflammatory genes including IL-6, IL-17A, and IL-1β in I3C-treated *Cr*-infected mice compared to *Cr*-infected mice fed on the control diet. However, our results on I3C/DIM’s effect on *Cr* growth and adhesion suggest an additional explanation for the observations in AhR knockout mice and support the concept that I3C may also act through mechanisms proximal to activation of host immune system to protect against *Cr* infection and inflammation.

Ingestion of dietary I3C reduced *Cr* levels in the feces and colon of mice in the present study. Furthermore, I3C reduced infection-induced colon hyperplasia on day 12 but not on day 21. Reduced colon hypertrophy correlated with a reduction in pro-inflammatory cytokine production, an increase in the expression of IL-10, and enhanced expression of the Treg marker Foxp3 as well as the cytotoxic T-cell markers CD8a, CD8b, and FasL. These data further support dietary I3C or cruciferous vegetable in moderating the impact of infectious colitis. *Cr* colonization elicits a robust T cell response in the colon [22,33]. The number of CD4^+^ T cells and CD8^+^ T cells was reported to peak on day 14 post infection [34,35]. The role of CD4^+^ T cells in immunity for *Cr* is well known [22], but there are relatively fewer studies on the role of cytotoxic T cell (CD8^+^ T cell) and Treg cell in *Cr* infection. Data presented here for cytotoxic T cell and Treg markers support the notion that there is also a role for I3C in modulating immune pathways related to these cells. At day 12 post infection, the observed increase in gene expression of CD8a, CD8b, and FasL mRNA indicates that the cytotoxic T cell pathway was upregulated in the colon of *Cr*-infected mice. More importantly, infected mice fed the I3C diet exhibited a higher expression of CD8a, CD8b, and FasL genes than infected mice fed on the control diet. These results indicate that I3C may enhance the cytotoxic T cell response and may be beneficial for protecting against colon inflammation and therefore carcinogenesis. Recent advances in cancer immunotherapy indicate that activating cytotoxic T cells may be an effective strategy for treating cancer [36] and I3C enhancement of cytotoxic T cell responses would complement such immune therapy.

Given the role of Treg in the resolution of inflammation, increased colonic expression of the Treg marker Foxp3 and IL-10 by dietary I3C may have contributed to a reduction in hyperplasia by dampening the pro-inflammatory response. These data are consistent with a previous study that documented AhR activation triggered Treg cell during DSS-induced colitis and support regulation of Treg as an additional mechanism by which I3C can protect against *Cr* infection. Additional studies are needed to fully elucidate the role of Treg in the protective effects of I3C. The role of Treg in colon cancer development remain paradoxical and both pro- and anti-tumorigenic roles have been reported. Recently, it was reported that tumor-infiltrating Foxp3 Tregs predict a favorable outcome in colorectal cancer patients. Moreover, to explain the paradoxical role, it was proposed that Treg regulation by the gut microbiome/bacteria may provide protection against colon cancer. Hence, up-regulation of Treg by I3C as observed in our system may be relevant in preventing colon cancer and warrants further investigation.

## 5. Conclusions

In summary, the data presented here supports a protective role of I3C in *Cr*-induced infection/inflammation and thus, by inference, may reduce the risk of CRC. In addition to immunomodulatory effects on *Cr* infections, I3C may also affect growth and adhesion of *Cr*. Our results also indicate regulation of cytotoxic T cell and Treg cells as additional molecular mechanisms that may be regulated by I3C which can lead to overall anti-inflammatory and anti-colon cancer potential.

## Figures and Tables

**Figure 1 nutrients-12-00917-f001:**
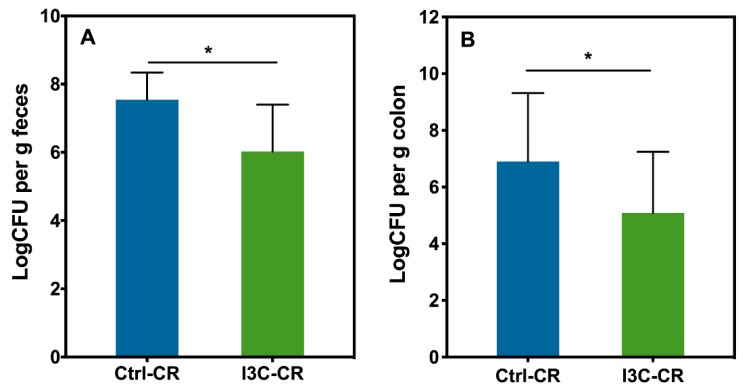
Effect of Indole-3-carbinol (I3C) on fecal and colonic *Citrobacter rodentium* (*Cr*) in mice. Mice were infected orally with approximately 1.0 × 10^10^ CFU of *C. rodentium*. (**A**): Fecal excretion of *Cr*. The feces of mice were collected on day 11 after infection and the Log cfu/g feces determined set; (**B**): Colonic colonization of *C. rodentium*. Mice were sacrificed on day 12 post infection and the Log cfu/g colon tissue was determined. Results were expressed as mean +/− SD (*n* = 8). * indicates significantly different from control (*p* < 0.05).

**Figure 2 nutrients-12-00917-f002:**
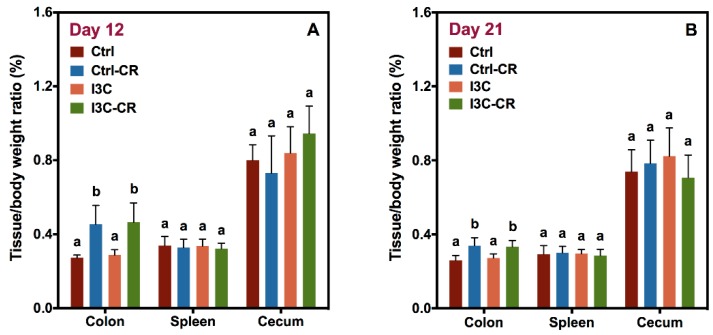
*Cr* infection resulted in an increase in colon weight on day 12 and 21 post infection but had no impact on spleen and cecum weight. Colon, spleen and cecum samples were collected and weighed on day 12 (**A**) and day 21 (**B**) after infection. Results were expressed as mean +/− SD (*n* = 8). Diet had no effect on tissue weights in either uninfected or *Cr*-infected mice. Significant differences (*p* < 0.05) between groups were identified by different letters.

**Figure 3 nutrients-12-00917-f003:**
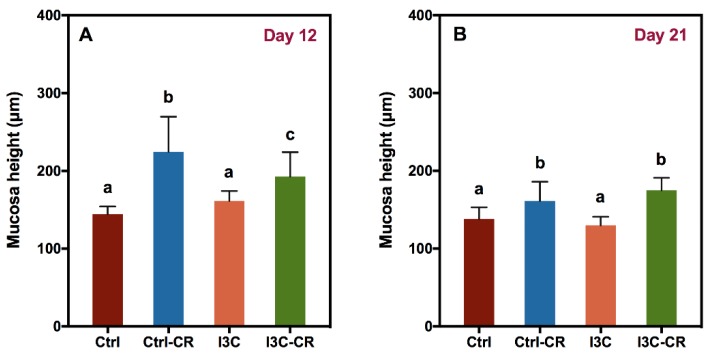
I3C reduces colonic hyperplasia at day 12 but not day 21 post infection. Colon tissue was collected on day 12 (**A**) and day 21 (**B**) after infection and processed for H&E staining. Mucosa height was measured on well-oriented crypts and 12 or more individual measurements were averaged for each mouse. Results were expressed as mean +/− SD (*n* = 8). Significant differences (*p* < 0.05) between groups are identified by different letters.

**Figure 4 nutrients-12-00917-f004:**
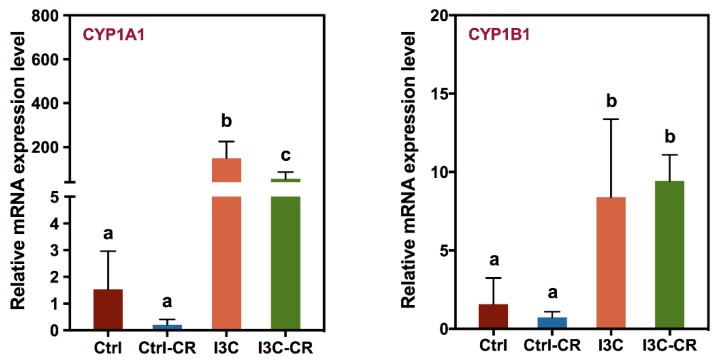
Xenobiotic metabolizing enzyme genes were upregulated in colons of mice fed the I3C diet. Total RNA isolated from colon tissues were harvested on day 12 after infection. The expression of cytochrome P450 1A1 (CYP1A1) and 1B1 (CYP1B1) enzyme genes were determined by using real-timeRT-PCR. Results were expressed as mean +/− SD (*n* = 8). Different letters indicated significant differences between groups for each tissue (*p* < 0.05).

**Figure 5 nutrients-12-00917-f005:**
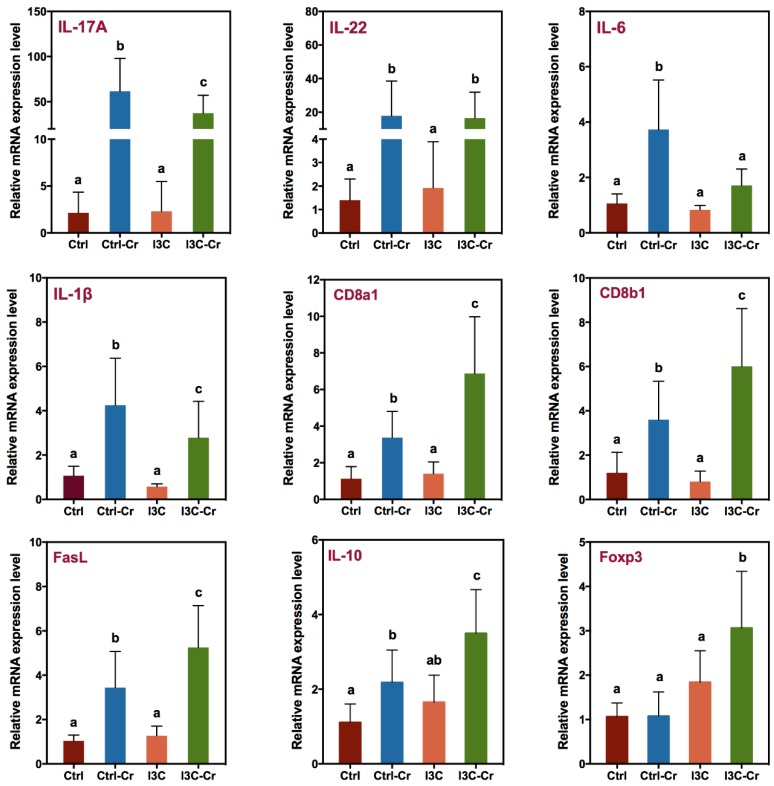
Effect of dietary I3C on *Cr*-infection-induced inflammation and immune cell markers gene expression in colon. RT-PCR was performed on total RNA isolated from colon tissues harvested on day 12 post infection. Results are expressed as the mean +/− SD fold-change (*n* = 8). Groups with different letters are significantly different. (*p* < 0.05).

**Figure 6 nutrients-12-00917-f006:**
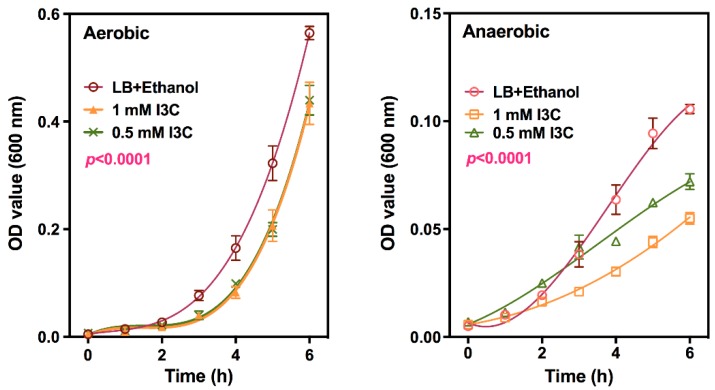
Effect of I3C on *Cr* growth in vitro. *Cr* (2 × 10^7^ cfu/mL in LB medium) were grown in the presence of carrier (ethanol), 0.5 or 1.0 mM I3C at 37 °C with shaking under aerobic or anaerobic conditions. OD 600 nm values were determined every hour after incubation by using microtiter plate reader. Data expressed as mean OD value ± SD (*n* = 8).

**Figure 7 nutrients-12-00917-f007:**
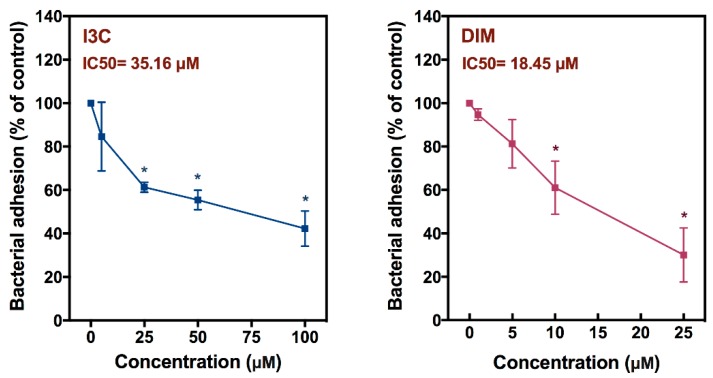
Effects of I3C and 3,3′-diindolylmethane (DIM) on adhesion of *Cr* to Caco-2 cell in vitro. Caco-2 cells were seeded in 24-well plates for 2 days and then co-cultured with the bacteria in the presence of I3C or DIM for 1 h. After incubation, the bacteria adherent to the cells were counted as described in the Materials and Methods section. Results expressed as mean +/− SD (*n* = 4) are shown as the percentage of *Cr* attachment relative to control cultures. * indicates significantly different from control (*p* < 0.05).

**Figure 8 nutrients-12-00917-f008:**
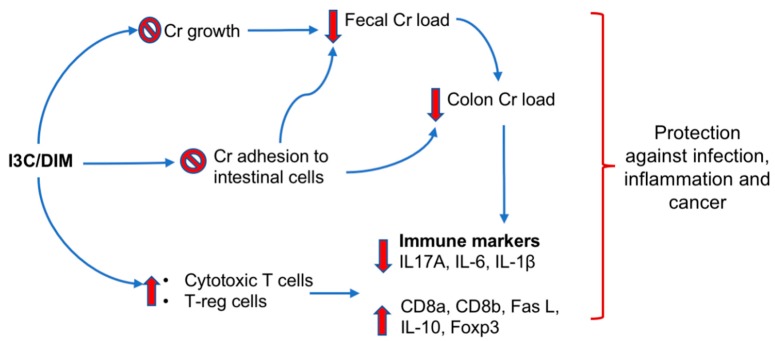
Working model for protective effects of I3C/DIM on infection, inflammation and cancer. I3C/DIM exert its protective effects through multiple mechanisms that include affecting bacterial growth, adhesion as well as immune responses mediated by AhR. Overall, the effects lead to decreased inflammation and protection against cancer.

**Table 1 nutrients-12-00917-t001:** The list of PCR primers used in this study.

Primers	Catalog Number	Primers	Catalog Number
TATA-box binding protein (TBP)	Mm00446971_m1	IL-1β	Mm00434228_m1
CYP1A1	Mm00487217_m1	CD8a1	Mm01182108_m1
CYP1B1	Mm00487229_m1	CD8b1	Mm00438116_m1
IL-17A	Mm00439618_m1	FasL	Mm00438864_m1
IL-22	Mm01226722_g1	IL-10	Mm00439614_m1
IL-6	Mm00446190_m1	Foxp3	Mm00475162_m1

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
