# Peer review of "Indole-3-Carbinol Inhibits Citrobacter rodentium Infection through Multiple Pathways Including Reduction of Bacterial Adhesion and Enhancement of Cytotoxic T Cell Activity"

_nutrients, 2020, doi:10.3390/nu12040917_

Round 1

Reviewer 1 Report

The authors of the manuscript "Indole-3-Carbinol inhibits Citrobacter Rodentium infection through multiple pathways............ Cytotoxic T Cell Activity"has done a good job of understanding the mechanism of protection by 13C against Cr but there are few fundamental questions to be answered:

  1. The authors should clarify the when the food with 13C was given and when the Cr was introduced and when the animals were sacrificed.
  2. The authors never showed any data of day 21 sacrifice. 
  3. The authors in discussion have said that the load of Cr decreases with 13C but in the figs its opposite.  If 13C does inhibit the attachment of Cr then the load should increase in feces and decrease in colon but it increases in both.
  4. Finally the authors are not clear about the final end point of the experiment.So with Cr infection of 20 days what is the final out come and how 13C is protective.
  5. According to authors 13C was given before the Cr treatment but in practicality its not the case and have the authors tried giving 13C post Cr treatment.

Author Response

Point-by-point responses

Reviewer 1:

The authors of the manuscript "Indole-3-carbinol inhibits Citrobacter rodentium infection through multiple pathways including reduction of bacterial adhesion and enhancement of cytotoxic T cell activity" has done a good job of understanding the mechanism of protection by 13C against Cr but there are few fundamental questions to be answered:

Point 1: The authors should clarify the when the food with 13C was given and when the Cr was introduced and when the animals were sacrificed.
Response: Thanks for reviewer’s suggestion. The corresponding descriptions were added into the Method section and highlighted in the manuscript. After a one-week acclimation period, we fed the mice experimental diet (with or without I3C) for two weeks prior to initiation of the Cr infection and for the remainder of the experiment. After two-week dietary intervention, the mice were infected by oral gavage with Cr, and then were sacrificed on day 12 and 21 after infection.

Point 2: The authors never showed any data of day 21 sacrifice.

Response: The tissue’s weight, colon H&E staining and the mucosa height on day 12 and day 21 after infection (Figure 2, S3, 3) were presented respectively in this study. The existing data showed I3C has no significant effects on the infected mice which were sacrificed on day 21 post-infection.

Point 3: The authors in discussion have said that the load of Cr decreases with 13C but in the figs its opposite. If 13C does inhibit the attachment of Cr, then the load should increase in feces and decrease in colon, but it increases in both.

Response: Thanks for the reviewer’s question. Figure 1A and 1B showed the numbers of Cr in feces on day 11 post-infection and in colon of infected mice on day 12 post-infection, respectively. Significant decreases in the Cr load were observed in the feces and the colon of mice fed I3C diet. We agree that the Cr load in feces might be higher in the infected mice fed I3C diet at some point in the beginning of the infection, which are needed to be evaluated in further study. However, the lower Cr loads on day 12 post-infection, when the mice start to clear the infection, likely reflects the effects of IC3 on both adherence and the immune response to the infection

Point 4: Finally, the authors are not clear about the final end point of the experiment. So, with Cr infection of 20 days what is the final outcome and how 13C is protective. According to authors 13C was given before the Cr treatment but in practicality it’s not the case and have the authors tried giving 13C post Cr treatment.

Response: We thanks the reviewer’s comments, the corresponding descriptions in Introduction and Methods sections were revised to make the experiment design clearer. In general, Cr colonization peaks on day 7 post-infection, and is completely cleared by 2-3 weeks post-infection. Our results showed that I3C treatment reduced Cr loads, attenuated the pro-inflammatory immune response and colonic hyperplasia on day 12 post-infection, indicating a protective effect of I3C against Cr-induced pathology. In addition, both control and I3C mice had very low levels of fecal Cr at day 20 post-infection and the degree of colonic hyperplasia at day 21 post-infection was the same between the treatments, indicating that the primary effect of I3C treatment was earlier in the infection cycle and did not affect the resolution phase. Therefore, no further analyses were performed on day 21 samples.

It is possible that feeding mice I3C may influence the outcome of a pre-existing infection. However, these experiments that would likely involve testing of I3C at both dietary and pharmaceutical does (i.e., not achievable in the diet) and are beyond the scope of the current study which focused on the effects of regular consumption of I3C from the diet on a Cr infection. 

In the revised manuscript,

Page 2, lines 37 to 39, the new text “Following oral administration, Cr initially colonizes in the caecum and then migrates to the colon with rapid expansion, the Cr excretion usually peak on day 7 post infection, and is gradually cleared by 21-28 days after infection.” was added in the Introduction section.

Page 5, lines 217 to 219, the new text “On day 20 post-infection, the number of Cr in feces of the infected mice were low, and there is no significant difference in fecal Cr excretion between the infected mice fed with control and I3C diet. (Figure S2)” was added in the Result section.

Reviewer 2 Report

Dear authors,

The paper is interesting, well written and also presents interesting results regarding Indole-3-Carbinol and colon health especially in the context of the importance of gut microbiota and general health.

Meanwhile the effects of infection by Citrobacter rodentium are less important or even not significant at day 21:

P6: “I3C treatment has no effect on the infection-induced increase in colon weight”

And P7: “I3C reduces colonic hyperplasia at day 12 but not day 21 post-infection”

Could you comment and/or add some results like in “Figure 1. Effect of I3C on fecal and colonic C. rodentium in mice” on day 21 if available?

Also, little oversight on p6:

On figure 1: « A » forgotten in graphic

A paper talks about possibilities of a toxicity of the supplement of the study (Indole-3-Carbinol) and I would like the authors to discuss this point regarding the dose they use and what they would advise for a human supplements (which already exist!):   

Fletcher AHuang HYu LPham QYu LWang TT. Reversible Toxic Effects of the Dietary Supplement Indole-3-Carbinol in an Immune Compromised Rodent Model: Intestine as the Main Target. J Diet Suppl. 2017 May 4;14(3):303-322.

"In this model, the calculated 75 mg equivalent consumed by the animals in this study would be much lower exposure concentration in the intestine as compared to a 200 or 400 mg commercial tablet. Hence, considering the adverse effect observed in this study, an immune-deficient/compromised population should be especially cautious when consuming commercial I3C tablet."...

"CONCLUSION In this study, we found intestinal damage occurred in an immune-deficient animal model that received I3C supplementation as low as 10 μmole/g diet. This is a concentration comparable to or lower than orally administered doses in humans (Reed, et al., 2006; Reed, et al., 2005). Our study demonstrated potentially toxic effects of I3C supplement that appeared to be specific to the gastrointestinal tract. This study should serve as a caution for use of I3C supplements in humans, especially in the immune-deficient/compromised population, such as cancer, transplant, and AIDS patients."

Best regards

Author Response

We thanks the reviewers suggestion. Please see the attachment.

Round 2

Reviewer 1 Report

The authors have given satisfactory answers the to question asked and made changes in the manuscript.